# Contrast normalization affects response time-course of visual interneurons

**Nadezhda Pirogova**[1,2]*, **Alexander Borst**[1]

**1** Department Circuits-Computation-Models, Max Planck Institute for Biological Intelligence, Planegg, Martinsried, Germany, **2** Graduate School of Systemic Neurosciences, LMU Munich, Planegg, Martinsried, Germany

* nadezhda.pirogova@bi.mpg.de

**Data Availability Statement:** Raw data from calcium imaging experiments, Python environment and code to replicate the figures, and the modelling code are available in the GitHub repository: https://github.com/nopirogova/paper_signal_time-course/
.

## Abstract

In natural environments, light intensities and visual contrasts vary widely, yet neurons have a limited response range for encoding them. Neurons accomplish that by flexibly adjusting their dynamic range to the statistics of the environment via contrast normalization. The effect of contrast normalization is usually measured as a reduction of neural signal amplitudes, but whether it influences response dynamics is unknown. Here, we show that contrast normalization in visual interneurons of *Drosophila melanogaster* not only suppresses the amplitude but also alters the dynamics of responses when a dynamic surround is present. We present a simple model that qualitatively reproduces the simultaneous effect of the visual surround on the response amplitude and temporal dynamics by altering the cells' input resistance and, thus, their membrane time constant. In conclusion, single-cell filtering properties as derived from artificial stimulus protocols like white-noise stimulation cannot be transferred one-to-one to predict responses under natural conditions.

## Introduction

Due to the complexity of the natural environment, the sensory inputs animals receive can vary by several orders of magnitude, ranging from the uniformity of an open field on an overcast evening to the stark visual contrast of forest trees on a sunny day. The spectrum such inputs can encompass is broad but individual neurons have only a limited response range to map these inputs onto, be it the graded membrane potential or spike rates. Therefore, the range of the responses has to be used efficiently to match the inputs [1]. For the visual system, that means adapting not only to the mean light intensity of each of the scenes but also to the fluctuations of these intensities around the mean, i.e. the image contrast. Neurons solve this problem by contrast adaptation: they dynamically adjust their contrast sensitivity to the statistics of the current environmental conditions [2]. This way, the sensitivity of the neuron increases at low contrast, making the neuron more responsive to small changes, and decreases at high contrast so that large changes in the stimulus do not lead to response saturation [3–6]. Thus, the amount of information about the stimulus contained in the response is maximized [1, 2, 7]. Contrast adaptation in the vertebrate retina has also been shown to induce changes in the dynamics of the signal, increasing the signal processing speed, making the signal more transient and, thus, improving the cell's ability to encode fast temporal changes [3–5].

**Funding:** This work was supported by the Deutsche Forschungsgemeinschaft (SFB 870) (https://www.dfg.de/) and the Max-Planck-Gesellschaft (https://www.mpg.de/de). The funders had no role in study design, data collection and analysis, decision to publish, or preparation of the manuscript.

The fruit fly *Drosophila melanogaster* is able to successfully navigate a variety of environments with variable visual statistics [8]. Motion vision is a critical sensory cue for the fruit flies' course control system [9], where self-motion is estimated from the optic flow to respond to different environments reliably and robustly [10]. *Drosophila*'s visual system is well-studied, with a large genetic toolbox allowing for interrogating individual neurons of interest. The visual system of *Drosophila* starts at the retina and comprises four sequential neuropils, namely, lamina, medulla, lobula, and lobula plate (Fig 1A). Visual signals are processed retinotopically in the fruit fly's optic lobe. From lamina cells onwards, processing runs in two parallel pathways: an ON pathway processes light increments, an OFF pathway processes light decrements [11–13]. Neurons in the medulla have been previously described as either transient or tonic, based on their filtering properties [14]. Medulla neurons themselves do not respond selectively to the direction of visual motion but form the main inputs to the first direction-selective cells, T4 in the ON pathway, and T5 in the OFF pathway [15–17]. Contrast information is crucial in supporting robust behavior under various conditions, and lamina neurons are vital for distributing contrast information across the ON and OFF pathways [18]. Contrast normalization is present in the early visual system of *Drosophila* and arises first in transient neurons of the medulla [19]: a high-contrast grating presented in the visual surround outside the neuron's receptive field (RF) suppresses the amplitude of the response to a local stimulus presented in the center of the cell's RF. Contrast sensitivity is usually measured as the response of the cell as a function of the local contrast (Fig 1B). When this measurement is taken with a grating

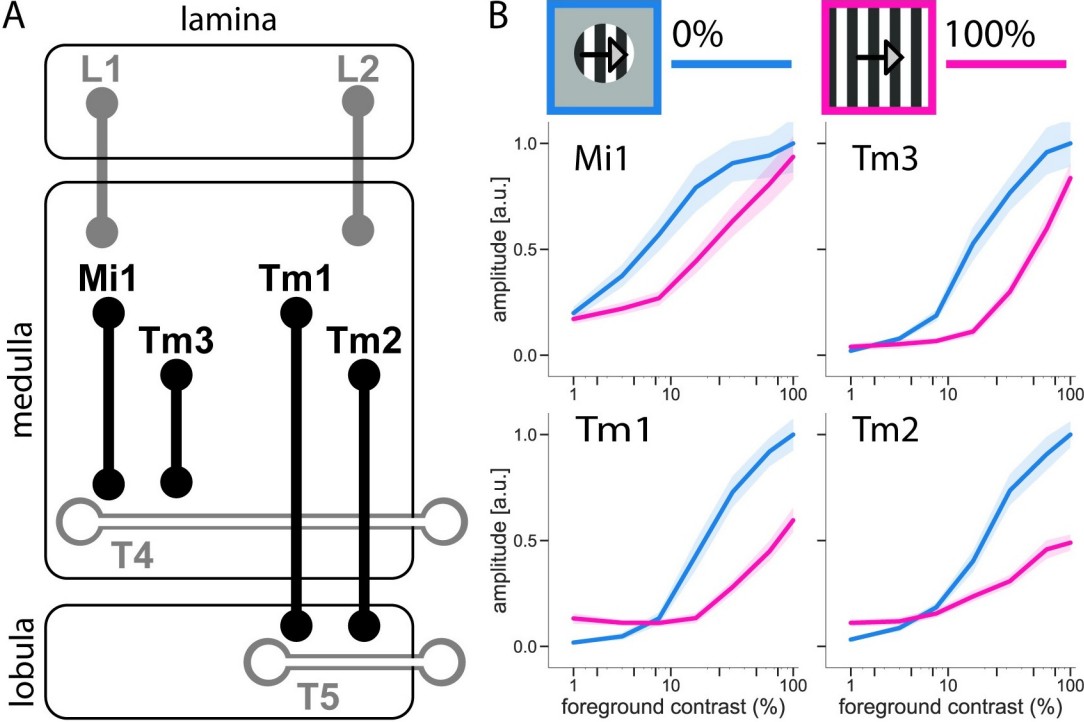

**Fig 1. Contrast normalization in early visual system of *Drosophila*.** (A) Schematic representation of early stages of the motion detection circuit. Highlighted are contrast normalization-exhibiting neurons that provide major input to T4 and T5 cells, the first direction-selective neurons in the ON and OFF pathways, respectively. (B) Contrast normalization experimental protocol and contrast tuning curves for different medulla neurons: Mi1 (n = 20 cells/5 flies), Tm3 (n = 21 cells/8 flies), Tm1 (n = 21 cells/7 flies), and Tm2 (n = 20 cells/6 flies). Shaded areas around the curves show bootstrapped 68% confidence intervals. Adapted from Drews et al. (2020) [19].

moving in the surround, contrast normalization shifts the response curve on a logarithmic contrast axis to the right. This shift adjusts the steep part of the neuron's response curve, i.e. where the cell is the most sensitive, to the prevalent contrast in the surround. Drews et al. (2020) [19] characterized the compressive, normalizing signal as fast, integrating spatially over a large area and deriving from neural feedback. Functionally, it significantly improves the robustness of motion detection in natural scenes [19]. The role of contrast adaptation in improving motion estimates in natural scenes was also confirmed in a recent study by Matulis et al. (2020) [20] who investigated contrast adaptation and its timescales in the early visual system of Drosophila.

To understand the mechanism by which T4 and T5 cells become selective for the direction of image motion, it is important to know the filtering properties of each of their input cells. To characterize the response dynamics of visual interneurons in *Drosophila*, the following artificial stimuli were commonly employed in the past: a) gratings with defined spatial wavelength and contrast moving at various velocities [15, 21–23]; b) moving edges of defined polarity [12, 24], i.e. either a dark edge on a bright background or the other way around; c) white noise stimuli consisting of statistically independent flickering pixels or bars [14, 25–27]; d) defined luminance pulses or steps placed in the center of the RF of the cell [23, 28]. Importantly in the present context, these stimuli differ from each other with respect to the amount of contrast present in the surround.

As was already shown, the contrast of the surround can strongly suppress the amplitude of the responses to the center stimulus [19]. In this paper, we asked whether contrast normalization in the *Drosophila* visual system has an influence not only on the amplitude of the response but also on its temporal dynamics, as was shown for the vertebrate retina [3–5]. This way, contrast normalization would affect the cell's filtering properties and, therefore, might influence its role in motion vision. We focused on four transient medulla neurons that have previously shown to exhibit contrast normalization and to provide major input signals to the first direction-selective cells in the ON and OFF pathways in *Drosophila* [16, 17]. By combining artificial stimuli widely used to characterize neuronal responses in *Drosophila melanogaster*, we aim to untangle the effects of the local and global stimulus profile on the cells' filtering properties.

## Materials and methods

### Data and code availability

Raw data from calcium imaging experiments, code to replicate the figures, and the modelling code are available in the GitHub repository: https://github.com/nopirogova/paper_signal_time-course/.

### Flies

Flies were raised and kept on standard cornmeal-agar medium on a 12h light/12h dark cycle at 25˚C and 60% humidity. The genetically-encoded calcium indicator GCaMP6f [29] was expressed using the Gal4-UAS system [30], resulting in the following genotypes:

L1>GC6f: *w+; VT027316-AD/UAS-GCaMP6f; R40F12-DBD/UAS-GCaMP6f*

Mi1>GC6f: *w+; R19F01-AD/UAS-GCaMP6f; R71D01-DBD/UAS-GCaMP6f*

Tm1>GC6f: *w+; R41G07-AD/UAS-GCaMP6f; R74G01-DBD/UAS-GCaMP6f*

Tm2>GC6f: *w+; R28D05-AD/UAS-GCaMP6f; R82F12-DBD/UAS-GCaMP6f*

Tm3>GC6f: *w+; R13E12-AD/UAS-GCaMP6f; R59C10-DBD/UAS-GCaMP6f*

The transgenic fly line driving split-Gal4 expression in L1 cells is courtesy of A. Nern, Janelia Research Campus; lines for Mi1 and Tm3 cells were generated and described in Strother et al. (2017) [23], for Tm1 and Tm2 in Davis et al. (2020) [31].

## Two-photon imaging

**Fly preparation.** The flies were taken 2–5 days after eclosion and prepared as previously described [15, 23]. In short, flies were anesthetized on ice, their backs, legs, and wings were fixed onto an acrylic glass holder, and the back of the head was exposed in a chamber filled with Ringer's solution. The cuticle behind the right eye was cut with a fine hypodermic needle and removed, along with muscles and air sacks, uncovering the optic lobe.

**Image acquisition.** Two-photon imaging [32] was performed on a custom-built microscope as described by Maisak et al. (2013) [15], controlled with the ScanImage software (version 5.1) in MATLAB [33]. Imaging was performed at an acquisition rate of 11.8 Hz with an image resolution of 128×128 pixels. As described by Arenz et al. (2017) [14], imaging stacks were automatically registered in a custom-written software to correct for the movement of the brain.

Regions of interest (ROIs) were drawn manually on the average raw image to extract responses of individual neurons. For ON pathway medulla cells, Mi1 and Tm3, ROIs were drawn in the medulla layer M10, for the OFF pathway medulla neurons, Tm1 and Tm2, in lobula layer Lo1, and for the lamina neuron, L1, in the medulla layer M1. Fluorescence changes (ΔF/F values) were then calculated using a standard baseline algorithm over the ROI [34].

## Visual stimulation

**Arena.** Stimuli were projected with 2 commercial micro-projectors (TI DLP Lightcrafter 3000) onto a custom-built cylindrical arena, as previously described by Arenz et al. (2017) [14]. Stimuli covered 180˚ in azimuth and 105˚ in elevation of the visual field of the fly.

The projectors had a refresh rate of 180 Hz (at 8-bit color depth), their medium brightness was set to the value of 50 on an 8-bit grayscale, corresponding to a medium luminance of 55 ±11 cd/m2. Stimuli were rendered using custom-written software in Python 2.7 and Panda3D framework.

**Gaussian noise stimulus.** The stochastic noise stimulus used to determine the location of a cell's receptive field was pre-rendered and generated as previously described in Arenz et al. (2017) [14]. Briefly, the 3-minute-long stimulus consisted of 64×52 pixels that covered the whole screen, one pixel corresponding to a visual angle of around 2.8˚. The intensities of each pixel were drawn from a Gaussian distribution at 100% contrast and low-pass filtered using a Gaussian window with a standard deviation of approximately 90 ms, corresponding to a binary noise with the temporal cut-off frequency of 1 Hz.

The response of an individual cell, as imaged within an ROI, was used to reconstruct the cell's spatiotemporal receptive field as described by Arenz et al. (2017) [14]. The obtained coordinates of its position on the screen were used to center the step stimulus as described below.

**Visual stimulation of the medulla neurons.** The coordinates of a cell's receptive field, obtained from its response to the Gaussian noise, were used to position the step stimulus. Before presenting the stimulus, we verified that the RF center was sufficiently distant from the border of the screen so that a significant part of the surround of the stimulus could be displayed.

The center of the stimulus comprised a 5˚ circular window, in which a 1-second-long step of luminance was presented. The contrast polarity of the step corresponded to the preference of the cell [25], i.e. for Mi1 and Tm3, the luminance of the center increased from 0% to 50%,

75%, or 100% during the step, and vice versa, the luminance of the center decreased from 100% to 50%, 25%, or 0% for L1, Tm1, and Tm2. The center window was surrounded by a 30˚ gray annulus (medium luminance), intended to cover the surround of the receptive field of the cell and prevent the stimulus surround from leaking into the cell's RF.

Beyond the annulus, one of the four surround conditions was shown, all with the same mean luminance. Each surround condition was shown 3 s before the luminance step in the center and remained on screen for 2 s after the step. In the cases when the surround condition was dynamic, the dynamics of the surround started 1 s before the luminance step in the center. The four surround conditions were as follows: 1. uniformly gray (contrast 0%); 2. 20˚-wavelength stationary grating (contrast 100%); 2. 20˚-wavelength grating moving at 1 Hz (contrast 100%); 4. stochastic stimulus, (here, a binary noise with the temporal cut-off frequency of 10 Hz, further properties as described in Gaussian Noise Stimulus).

All stimuli were repeated 3 times in randomized order to prevent adaptation to any stimulus features.

## Data analysis

Data analysis was performed offline using custom-written routines in Matlab and Python 2.7 and Python 3.7 (with the SciPy and OpenCV-Python Libraries).

**Data evaluation.** Relative fluorescence change (ΔF/F) was calculated using a standard baseline algorithm over an individual ROI as described in Jia et al. (2011) [34]. Briefly, raw signal was smoothed with a Gaussian window with FWHM of 1 s, the minima in a 90 second-long sliding window were extracted, and the trace was smoothed with a Gaussian window with FWHM of 4 min, resulting in the dynamic baseline, $F_0$. To better compare the time-course of the signals to different stimuli, response curves were also normalized to their maximum, i.e. divided by the peak value reached during the stimulation period.

To filter out the cells for which the variance in responses between trials was caused by movement artifacts, a signal-to-noise ratio (SNR) criterion was introduced. Here, only the recordings, in which the inter-trial variance was smaller than the average cell response, were taken, i.e. the standard deviation of the averaged signal had to be at least 115% of the mean standard deviation over trials. On average, over 90% of all cells measured passed the SNR criterion.

**Data visualization.** For all experiments, responses for each cell were averaged over trials, normalized to the cell's maximum, and further averaged over the cells. Additionally, to illustrate the temporal dynamics of the responses, responses of each cell type for every condition were normalized to the condition's maximum for the respective cell type.

To visualize response traces when a cell responded identically to several stimuli, an artificial gap was introduced by offsetting one of the responses vertically. In these cases, the figure legend specifies for which conditions the offset was introduced.

**Statistical tests.** Shaded areas around the response curves and error bars show bootstrapped 68% confidence intervals around the mean (estimated as corresponding distribution percentiles after resampling the data 1,000 times). All statistical tests were two-tailed and performed at a 5% significance level. Sample sizes are given in each figure legend. The experimenters were not blinded to genotypes or conditions during data gathering and analysis.

## Modelling

The model comprised three stages through which an input, a step function, was sequentially processed. The input was a 1 s pulse at either full amplitude or an amplitude of 0.2.

The processing cascade was as follows: a (1) band-pass filter, followed by either a (2.1) static or a (2.2) dynamic nonlinearity, followed by a (3) low-pass filter.

(1) The first stage of the model, a band-pass filter, simulated the response of a cell membrane to a luminance step received in the center of its RF. The band-pass filter was constructed as a combination of a low- and a high-pass filter (LP and HP) with $\tau_{LP}$ = 200 ms and $\tau_{HP}$ = 300ms.

(2) At the second stage, the response was transferred through a divisive nonlinearity, as often used to describe "contrast normalization" i.e. the saturating contrast dependency.

(2.1) The stationary nonlinearity was constructed as follows:

$$y(t) = \frac{x(t)}{x(t) + k},$$

where $x$ was the input and $y$ the output signal amplitude, and $k$ the parameter controlling the amount of saturation. $k$ values of 0.2 and 1.0 were used to simulate the static and dynamic surround experimental conditions, respectively.

(2.2) The dynamic nonlinearity was constructed as follows:

$$y(t) = \frac{x(t - \Delta t)E_{exc} + y(t - \Delta t)C/\Delta t}{x(t - \Delta t) + k + C/\Delta t},$$

where $x$ was the input and $y$ the output signal amplitude, $E_{exc}$ was set to 1.0, $C/\Delta t$ to 100.0, and $k$ was the parameter controlling the amount of saturation. $k$ values of 0.2 and 1.0 were used to simulate the static and dynamic surround experimental conditions, respectively.

This corresponds to:

$$V(t) = \frac{g_{exc}E_{exc} + V(t - \Delta t)C/\Delta t}{g_{exc} + g_{leak} + C/\Delta t},$$

where $g_{exc}$ is the conductance of the excitatory current, $V$ is the membrane potential, $E_{exc}$ is the reversal potential of the excitatory current, $C$ is the membrane capacitance, $\Delta t$ is the time-step, and $g_{leak}$ is the conductance of the leak current.

For an electrically passive membrane, the following equation states that the sum of all currents across the membrane equals zero:

$$-C\frac{dV(t)}{dt} = g_{exc}(V(t) - E_{exc}) + g_{leak}(V(t) - E_{leak}).$$

(3) At the last stage, to account for the calcium indicator (GCaMP6f) dynamics, the signal was processed through a low-pass filter with $\tau_{Ca}$ = 200 ms.

Finally, the responses were normalized as described in "Data Evaluation" so that the model output was comparable to the experimental data.

## Results

We performed *in vivo* 2-photon calcium imaging from axon terminals of Mi1 and Tm3 neurons in layer M10 of the medulla, and from axon terminals of Tm1 and Tm2 neurons in layer Lo1 of the lobula. These recording sites correspond to the locations where the respective neurons synapse onto the dendrites of the first direction-selective cells: Mi1 and Tm3 onto T4 cells, and Tm1 and Tm2 onto T5 cells (see Fig 1A).

As we were interested in the effects that contrast normalization has on single cells, our stimulus protocol included a stochastic noise stimulus to determine the location of the cell's RF.

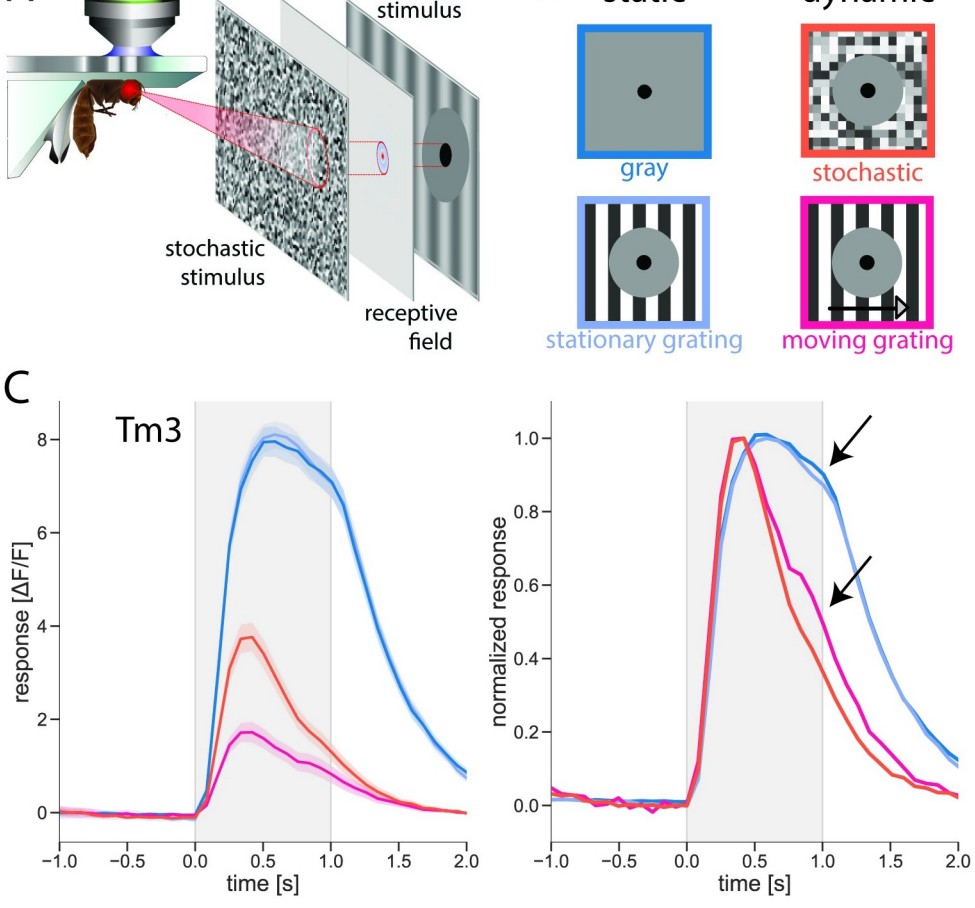

**Fig 2. Dynamic surround affects response amplitude and kinetics.** (A) Experimental procedure: (1) white noise stimulus, (2) receptive field (RF) reconstruction from single-neuron calcium signals, (3) experimental stimuli centered on RF. (B) Stimulus protocol. Luminance step in RF center with 4 surround conditions. (C) Average Tm3 response to luminance step in RF center with gray, stationary, moving grating, and stochastic stimulus surround; n = 23 cells/6 flies. Luminance step happened during the gray-shaded period. Left: Amplitudes of cell responses. Shaded areas around the curves show bootstrapped 68% confidence intervals. Right: Kinetics of cell responses. Responses during each condition are normalized to the condition's maximum. Artificial gap is created between responses to static conditions for easier visualization. See also S1 Fig.

This location was then used to project further stimuli to specific screen locations relative to the cell's RF (Fig 2A).

All of our stimuli followed the same pattern: a 1-second luminance step with a 100% amplitude was shown in a 5˚ circular window positioned in the center of the cell's receptive field. This step consisted of a luminance increase from 0 to 1 for Mi1 and Tm3, and a luminance decrease from 1 to 0 for Tm1 and Tm2. The window was surrounded by a 30˚ annulus at an intermediate luminance level that covered most, and often the entirety, of the cell's receptive field. The annulus ensured that the surround of the stimuli was not spilling over into the receptive field of the cell avoiding a direct stimulation of the neuron. The size of 30˚ of the annulus was based on the average full width at half maximum (FWHM) for the imaged cells: this value (rounded to the nearest integer) was previously determined to be 29˚ for Mi1 cells, 12˚ for Tm3 cells, 27˚ for Tm1 cells, and 31˚ for Tm2 cells [14]. Covering the rest of the screen (hereafter referred to as "surround"), sparing the center and the annulus, one of four different

surround stimuli was presented (Fig 2B): (i) uniformly gray, i.e. medium luminance, (ii) stationary grating at full contrast; (iii) 20˚-wavelength full-contrast grating moving at 1 Hz; (iv) full-contrast stochastic binary pixel noise. This selection of stimulus conditions ensured that we had both a static (conditions i and ii) and a dynamic (conditions iii and iv) surround. The two different dynamic surround conditions further differentiated between temporal dynamics containing motion (moving grating in condition iii) and no motion component (stochastic noise in condition iv).

## Effect of visual surround on response amplitude and kinetics

First, we wanted to probe the cells of interest to determine which features of the stimulus surround affected the response of the cell to stimulation of its receptive field center. To this end, our stimulus consisted of a luminance step in the center and one of the four conditions in the surround. Fig 2C shows responses of Tm3 neurons to the four stimulus conditions. For all the surround conditions, the cells responded to the luminance step with a fast signal increase that began decaying while the step was still present. However, the amplitudes of the response and the kinetics of the signal decay varied depending on the dynamics of the surround. Stationary grating in the surround had no visible effect on the signal, making the responses to the two static surround conditions, uniformly gray and stationary grating, virtually identical (Fig 2C). In the presence of a dynamic surround, however, the amplitude of the response was dramatically reduced. For a stochastic noise surround, the maximum signal amplitude was significantly reduced, only reaching 48% of the peak response value observed in the static surround condition (for precise values and statistical tests, see S3 Fig). When a moving grating was presented in the surround, the signal was suppressed even more strongly, with the maximum amplitude of the response reaching only 24% of the peak response value observed for the static surround condition.

In addition to suppressing the amplitude, the dynamic surround affected the kinetics of the responses (Fig 2C, right panel). As best seen after normalizing cell responses to their peaks, Tm3 responses to the two static conditions had virtually identical kinetics, with a rather slow decay of the signal during the luminance step. In the presence of a dynamic surround, however, responses decayed visibly faster and, by the end of the luminance step (arrows in Fig 2C), decreased to 36% and 49% of their peak amplitudes in the stochastic noise and moving grating surround conditions, respectively. These values correspond to a significantly stronger decay in the signal, when compared to the response in the presence of a static surround that decayed only to 86% of its peak value (for precise values and statistical tests, see S3 Fig). Interestingly, the temporal profiles of the responses in the two conditions with dynamic surround were virtually identical, as were the signals in the two static surround conditions.

These phenomena, i.e. response amplitude suppression and faster response decay caused by the dynamic but not the static surround, as well as extremely similar temporal profiles of the responses within the two static and the two dynamic surround conditions, were also observed in Mi1 (S1 Fig). Curiously, in the specific case of Mi1, a certain level of response was present already before the luminance step and was subsequently suppressed by the dynamic surround. We conclude that dynamic, but not static, surround has an effect on both the response amplitude and its kinetics.

Because of the high level of similarity of response time courses within the static and dynamic surround conditions, and as we were interested specifically in the effect of contrast normalization on the kinetics of the signal, we focused on one stimulus per condition category. Thus, in further experiments, we used the uniformly gray surround to represent the stationary and the moving grating for the dynamic surround condition.

### Response kinetics of transient medulla and transmedulla neurons

With this stimulus protocol, we probed the other cells, i.e. Mi1, Tm1, and Tm2, that had previously shown to exhibit contrast-normalizing properties [19]. Here, a similar picture arose (Fig 3, also see S4 Fig for extended traces). Firstly, all the cells tested responded to the luminance step in the center of their receptive field, regardless of the surround. Contrast normalization had a dramatic effect on the amplitude of the cells' responses, strongly decreasing the size of the response in the presence of moving grating in the surround (Fig 3A). Here, the response of Tm3 was suppressed the most: under dynamic surround conditions, the maximum amplitude of the response corresponded to 24% of the maximum reached when presented with a static surround. In comparison to the uniformly gray surround condition, the peak amplitude in the dynamic surround condition was also significantly decreased for the other cell types, constituting 67% for Mi1, 59% for Tm1, and 48% for Tm2 cells (for precise values and statistical tests, see S3 Fig).

The dynamic surround also had a pronounced effect on the temporal profile of the responses (Fig 3B), with the signal decaying faster when a moving grating was present in the surround. Numerically, when comparing the level of signal decay reached by the end of the

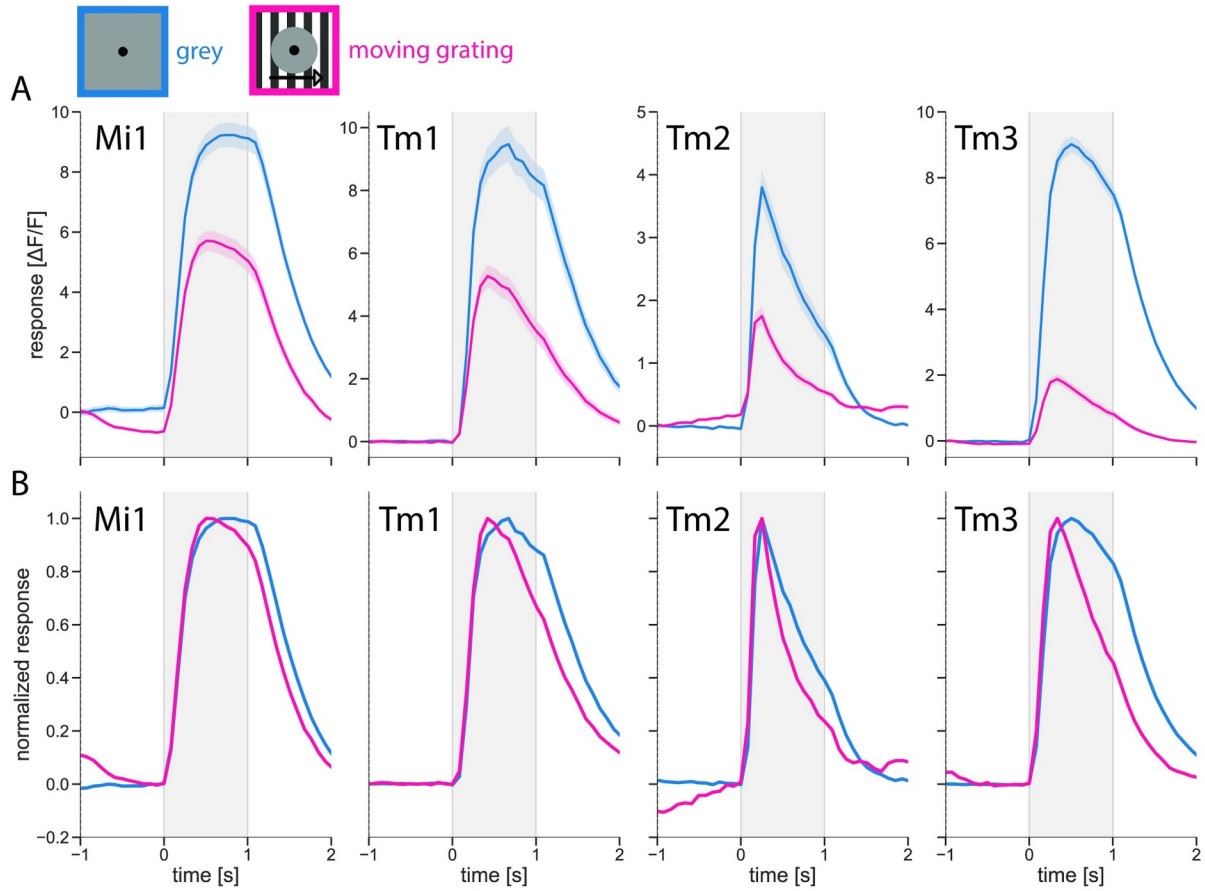

**Fig 3. Contrast normalization affects response amplitude and kinetics.** Average responses of contrast normalization-exhibiting neurons to luminance step in the RF center with gray and moving grating surround. Mi1 (n = 98 cells/25 flies); Tm1 (n = 24 cells/9 flies); Tm2 (n = 22 cells/7 flies), and Tm3 (n = 65 cells/16 flies). Mi1 and Tm3 datasets include the data from Figs 2 and 5. Luminance step happened during the gray-shaded period. (A) Amplitudes of cell responses. Shaded areas around the curves show bootstrapped 68% confidence intervals. (B) Kinetics of cell responses. Responses during each condition are normalized to the condition's maximum.

luminance step in the two conditions, Mi1 response decreased to 89% of its own maximum response amplitude in the dynamic condition, corresponding to a significantly stronger change than in the static surround condition, that decreased to 98% (for precise values and statistical tests, see S3 Fig). For the rest of the cells the signal also decayed significantly stronger in the presence of a moving grating than when a gray surround was present. The decay values for the dynamic and the static surround constituted, respectively, 67% and 88% for Tm1, 23% and 38% for Tm2, and 44% and 82% for Tm3 cells.

We conclude that, in all the transient medulla cells that had previously been reported to exhibit contrast normalization, the dynamic surround not only suppresses their response amplitude but also has a pronounced effect on their temporal profile.

## Modelling effects of dynamic surround

How can we explain that a moving stimulus in the surround, outside the receptive field of a neuron, affects both the amplitude and the time-course of the response? Classically, the phenomenon of contrast normalization is attributed to a divisive nonlinearity. The saturating contrast dependency of the response to a stimulus within the receptive field is well described by the following formula [2]:

$$y(t) = \frac{x(t)}{x(t) + k} \tag{1}$$

with $x$ being the input and $y$ the output signal amplitude, and $k$ the parameter which controls the amount of saturation: for small values of $k$ with respect to $x$, $y$ strongly saturates with increasing $x$ (Fig 4A). This corresponds to the situation of a static surround. Conversely, for $k$ being large with respect to $x$, $y$ grows in a rather linear way with increasing $x$. This corresponds to the situation where a grating is moving in the surround.

Consider the following signal processing cascade (Fig 4A). It consists of a band-pass filter (Fig 4A, left), the above-mentioned nonlinearity (Fig 4A, middle), and a final low pass filter (Fig 4A, right) to account for the calcium indicator dynamics. The nonlinearity is shown for two values of $k$, 0.2 (blue, representing static surround) and 1.0 (red, representing moving surround). If we stimulate the cell with a 1s pulse of light (Fig 4B, left), the signal amplitude will be different after the nonlinearity, depending on whether the surround is static or moving (Fig 4B, middle). However, the shape of the signal is also slightly different, as can be seen after the output signals are normalized to their maximum amplitude (Fig 4B, right). Thus, a stationary nonlinearity of the kind shown in formula (1) will affect not only the signal amplitude but also the signal dynamic.

If this explains the phenomenon shown in Fig 3, we should obtain the same signal amplitude and time-course for a static surround that we observed for a moving surround by simply reducing the stimulus amplitude accordingly (Fig 4C). A stimulus amplitude of 0.2 for a static surround resulted in a signal that is identical in amplitude and shape to the one resulting from a stimulus amplitude of 1.0 for a moving surround (Fig 4C, middle to right).

There is, however, another explanation for the phenomenon taking into account the biophysics of a neuron. The following equation describing an electrically passive membrane states that the sum of all currents across the membrane equals zero:

$$-C\frac{dV(t)}{dt} = g_{exc}(V(t) - E_{exc}) + g_{inh}(V(t) - E_{inh}) + g_{leak}(V(t) - E_{leak}) \tag{2}$$

Here, V(t) is the membrane potential, C the membrane capacitance, $g_{exc}$, $g_{inh}$, $g_{leak}$ are the conductances and $E_{exc}$, $E_{inh}$, $E_{leak}$ the reversal potentials of the excitatory, the inhibitory and

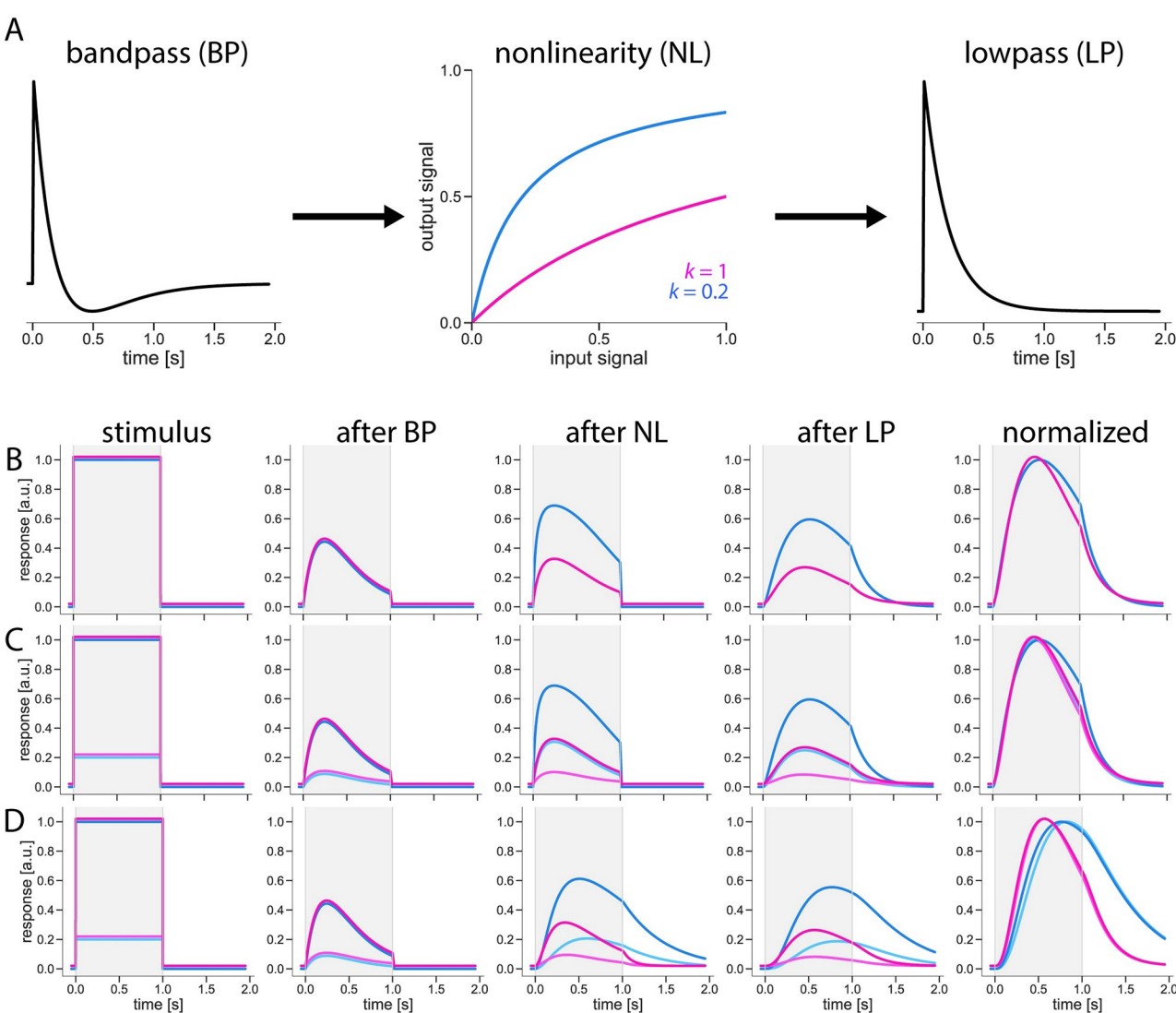

**Fig 4. Model with dynamic nonlinearity reproduces dynamic surround effect on response amplitude and kinetics.** (A) Model schematic: input step is sequentially passed through band-pass filter (BP), nonlinearity (NL), and low-pass filter (LP). NL with k = 1 corresponds to moving grating surround condition, k = 0.2 to gray surround condition. (B) Model cascade with stationary nonlinearity. (C) Model cascade with stationary nonlinearity and varying input amplitudes. (D) Model cascade with dynamic nonlinearity and varying input amplitudes. (B-D) For easier visualization, an artificial gap is created between model responses to moving grating and gray surround conditions.

the leak currents, respectively. The excitatory conductance is controlled by the signal within the neuron's receptive field, the inhibitory conductance is controlled by the stimulus presented within the background, outside the neuron's receptive field. Under steady-state conditions, i.e. $dV(t)dt = 0$, assuming $E_{inh} = E_{leak}$ (silent or shunting inhibition) and considering V(t) relative to $E_{leak}$ (i.e. $E_{leak} = 0$), this equation becomes:

$$V(t) = \frac{g_{exc}}{g_{exc} + g_{inh} + g_{leak}} E_{exc} \tag{3}$$

With $g_{exc}$ being the signal driving V(t), the correspondence to Eq (1), i.e. contrast normalization, is obvious: $g_{inh}$ becomes the factor controlling the amount of saturation. Under non-

steady-state conditions, however, $g_{inh}$, together with C, affects the membrane time-constant and thus the dynamic of the membrane voltage. In other words: taking into account the biophysics of a neuron's membrane, contrast normalization will alter the dynamic of the output signal, in addition to the effect described above, by altering the membrane time constant.

Rewriting Eq (2) as a difference equation results in the following equation:

$$V(t) = \frac{g_{exc}E_{exc} + V(t - \Delta t)\frac{C}{\Delta t}}{g_{exc} + g_{inh} + g_{leak} + \frac{C}{\Delta t}}$$

(4)

With the capacitive current being small relative to the leak and excitatory current, Eq (4) degenerates to Eq (3), i.e. it describes the membrane voltage under steady-state conditions.

Can we still observe the same time-course for a small stimulus amplitude with a static surround and a large stimulus amplitude with a moving surround? As is shown in Fig 4D, the answer is 'No'. Given a sufficiently large value of membrane capacitance, the two conditions cannot be interchanged with any combination of stimulus amplitudes: for both stimulus amplitudes, the responses with a moving surround are always faster than the responses with a static surround.

## Dynamic nonlinearity in the model recapitulates temporal effects of contrast normalization

In order to test which of the two explanations describes our data best, i.e. whether contrast normalization also affects the membrane time constant, we repeated the experiments on Tm3 and Mi1 cells, this time using three different stimulus amplitudes: 50%, 75%, and 100% (Fig 5A).

With decreasing input amplitude, the amplitude of the response also decreased, scaling, however, in a non-linear way (Fig 5B and 5C). In Tm3, inputs of 75% and 100% in the dynamic surround condition yielded virtually identical response amplitudes, while the amplitude of the response to a 50% luminance step was significantly lower (Fig 5B, left). Mi1 exhibited similar behavior in the static surround condition (Fig 5C, left).

However, inspection of the temporal profile of the responses (Fig 5B and 5C, right) reveals that the kinetics of the signals remained the same within the static and the dynamic surround conditions, independent of the input amplitude, with responses decaying visibly faster in the presence of a dynamic surround. Here, by the end of the luminance step, Mi1 responses to 100%, 75%, and 50% input amplitudes decayed similarly to 91%, 91%, and 86% of their respective maximum if a moving grating was present in the surround (for precise values and statistical tests, see S3 Fig). In contrast, if the surround was uniformly gray, the responses to all three input amplitudes also decayed similarly to each other, to 97–99% of the respective maximum, significantly less than the decay seen in the dynamic surround condition. Similarly, for Tm3, the signal also decayed equally in response to the 100%, 75%, and 50% input amplitudes, reaching, respectively, 52%, 42%, and 40% of its maximum in the dynamic surround condition and 80–82% if the surround was static.

Additionally, we used the same stimulus protocol and imaged the response of the lamina neuron L1 (S2 Fig), previously shown not to exhibit contrast normalizing properties [19]. Here, the amplitude of the response showed a clear dependence on the amplitude of the input step while the temporal kinetics of the signal remained the same, independent of the properties of the surround or the amplitude of the input.

Taken together, our results demonstrate that the amplitude of the input stimulus does not have a pronounced effect on the temporal properties of the response. The results also show that the differences in the signal time course are purely linked to the presence or absence of a

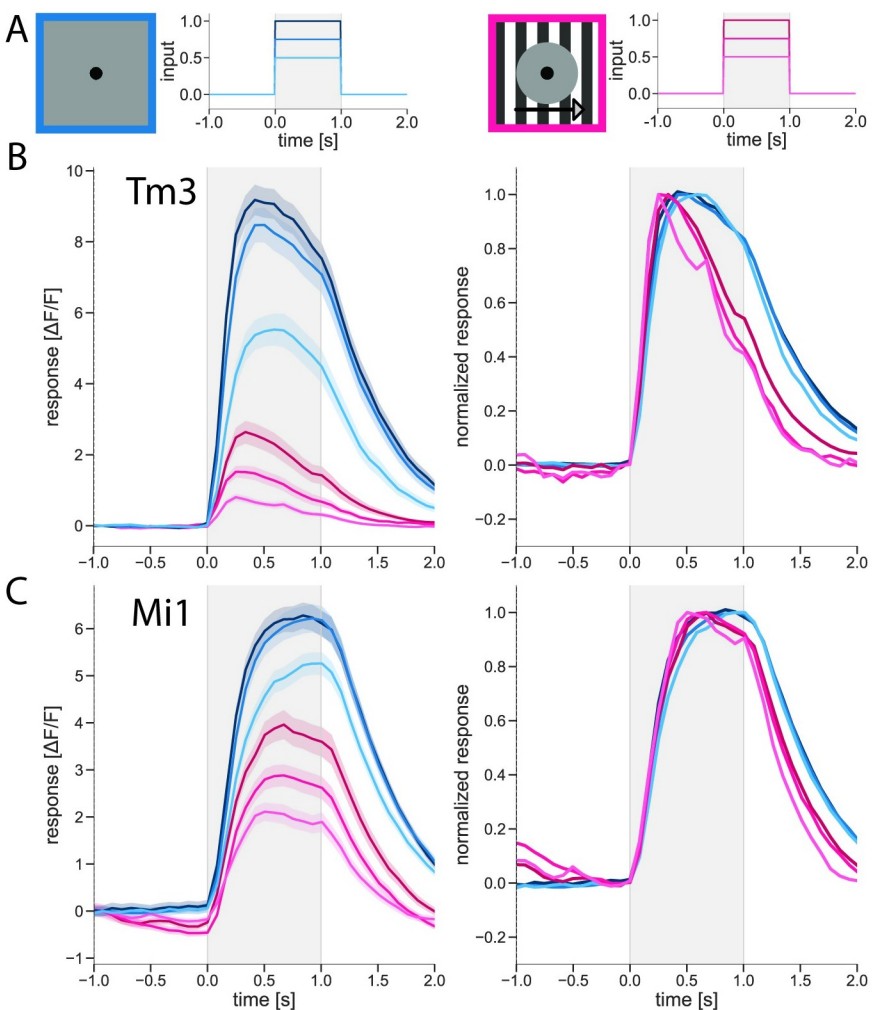

**Fig 5. Luminance step amplitude has no effect on response kinetics.** (A) Spatial and temporal stimulus profile. Left: 3 luminance step amplitudes with gray surround. Right: 3 luminance step amplitudes with moving grating surround. (B) Tm3 responses to 50%, 75%, and 100% luminance steps with gray surround; and to 50%, 75%, and 100% luminance steps with moving grating surround; n = 21 cells/4 flies. Luminance steps happened during the gray-shaded period. Left: Amplitudes of cell responses. Shaded areas around the curves show bootstrapped 68% confidence intervals. Right: Kinetics of cell responses. Responses during each condition are normalized to the condition's maximum. An artificial gap is created between responses to 100% and 75% steps for easier visualization. (C) Mi1 responses to 50%, 75%, and 100% luminance step with gray surround; and to 50%, 75%, and 100% luminance step with moving grating surround; n = 26 cells/8 flies. Luminance step happened during the gray-shaded period. Left and right as in (B). See also S2 Fig.

dynamic surround. Therefore, the change in the response dynamics cannot be explained as resulting from a static saturation nonlinearity. Conversely, the results are in line with changes in the membrane time constant that depend on the properties of the surround, thus supporting a model with a dynamic nonlinearity (Fig 4D).

## Discussion

Characterizing the temporal filter properties of visual interneurons, we have shown that contrast normalization has a large influence not only on the amplitudes of the cells' responses but also on their temporal dynamics. This effect was present in all medulla neurons that had

previously been shown to possess contrast normalization properties (Fig 3) and was exerted only by dynamic, but not by stationary stimuli present in the visual surround of the cells. We considered two models to explain this effect, one with a stationary, the other with a dynamic nonlinearity (Fig 4). By demonstrating that the temporal profile of the response was only influenced by the stimulus surround (Fig 5) and was not a simple consequence of response saturation, we could exclude the model with a stationary nonlinearity. The results were in line with our hypothesis that contrast normalization affects the membrane time constant. The results, thus, favor the model with a dynamic nonlinearity.

In Drews et al. (2020) [19], the presence of contrast normalization in *Drosophila*'s optic lobe and its suppressive effect on the gain of the response has been established. Here, we demonstrate the effect of contrast normalization on the temporal kinetics of the responses. Our results parallel the findings from the vertebrate retina, where cells' processing has also been shown to speed up, leading to their responses becoming more transient at higher contrasts [3–5]. At the same time, our findings seem to be, at first sight, hard to reconcile with a study by Matulis et al. (2020) [20] who investigated temporal contrast adaptation in the *Drosophila* optic lobe. They found different cells to be affected by the visual contrast than the current study as well as what was described in Drews et al. (2020) [19]. Furthermore, in electrophysiological recordings, contrast adaptation had no effect on the temporal dynamics of the signal, while, in two-photon calcium recordings from the same cells' dendrites, higher contrasts were found to slow the signals down—a finding that is opposite to the results from the vertebrate retina [3–5] and from our study. Whatever the explanation of the latter differences between voltage and calcium signals may be, these discrepancies may be partly attributed to the differences in the stimuli applied. In order to study contrast adaptation, i.e. the change in the cell's signal in response to the contrast of the entire visual scene, Matulis et al. (2020) [20] used full-field stimuli consisting of stochastic binary noise that switched between periods of low and high contrast. Hence, the change in stimulus contrast applied to both the cells' receptive field as well as the overall surround simultaneously. Our goal was to study contrast normalization, i.e. the influence of the surround stimulus on the response of the cell to the stimulus within its receptive field. Therefore, we as well as Drews et al (2020) [19] designed the stimulus such as to stimulate the cells' receptive field and the surround separately. Taken together, for most of the cells, the effects of contrast normalization on the amplitude and the dynamics of the response to stimuli delivered within the receptive field of the neurons, as found in Drews et al. and in this paper, seem to be distinct from the effects of contrast adaptation, as found by Matulis et al.(2020) [20]. For Mi1, however, Matulis et al. (2020) [20] also used local stimulation as a part of the stimulus set. The local stimulus elicited contrast adaptation but had no effect on the temporal profile of the response [20] so the discrepancy in the stimulus protocol cannot fully account for the difference in the results.

## Implications for characterizing filter properties of visual interneurons

In order to characterize the filter properties of single cells, a variety of different artificial stimuli are typically used, amongst them stochastic pixel noise, bars, and gratings, to name only a few. These stimuli offer numerous advantages for systematic studies: they are easy to parametrize by their mean luminance, contrast, and spatial wavelength, and the resulting response is readily evaluated in terms of the cell's impulse response, receptive field size, and frequency spectrum. However, the visual system has evolved as an adaption to complex natural images. Therefore, the results obtained from the simplified artificial stimuli may not be transferable to a cell's responses to naturalistic stimuli. Unlike natural environments, many of the stimuli used to characterize response properties of visual interneurons are tailored to the receptive

field of the cell under study. However, as we have shown above, the presence or absence of dynamic stimuli in the visual surround, far outside of the receptive field of a cell, has a pronounced influence not only on the response amplitude but also on the temporal response properties of the cell.

This has immediate implications for the interpretation of results obtained from visual interneurons in general, and for neurons involved in motion vision in particular. Here, the various models proposed to account for this computation [35, 36], although different in detail, all share the principle that a direction-selective output is achieved by a non-linear operation performed on differentially filtered signals derived from adjacent image points. In the fly visual system, T4 and T5 cells are known to be the first direction-selective neurons [15]. While contrast normalization has indeed been demonstrated in a number of columnar neurons providing input to T4 and T5 cells [19], their specific contributions to motion vision were derived from their temporal filtering properties which, in turn, were determined using stochastic pixel noise [14]. However, depending on the specific cell under study, the temporal response properties might be quite different whether the cell is stimulated by a full-field grating or by a local contrast step. To avoid this dilemma and in order to feed the model with faithful signals, input neurons should ideally be measured under stimulus conditions identical to those used to characterize the motion-sensitive neurons. Interestingly, this exact path has been chosen in a recent study on T4 cells [37].

## Functional consequences of contrast normalization

We focused on the visual interneurons with band-pass filtering properties [14], that provide excitatory input onto direction-selective T4 and T5 cells [17], exhibit contrast normalization [19], and are often used as input signals of various motion detector model simulations [14, 26]. Incorporating spatial contrast normalization into correlation-based models of motion vision has already been demonstrated to drastically improve the models' performance [19]. Here, we show that, in addition to the suppressive effect on the amplitude of the response, contrast normalization also alters the temporal dynamics of the signal and, thus, the filtering properties of the neurons. This change in the time course of the inputs has potential implications for the temporal dynamics of T4 and T5 neurons by adjusting the cells' speed tuning to the prevailing speed in the surround. To determine the influence that visual surround has on temporal filtering properties and velocity tuning of the neurons, the motion vision circuit needs to be scanned with a set of dynamic global stimuli at varying frequencies.

Contrast normalization does not alter the amplitude and the temporal dynamics of all the cells it affects by an equal amount. Consequently, the relative contribution of the different inputs to the T4 and T5 cells will vary depending on the global structure of the stimulus. In the ON pathway, in the presence of a global visual surround, Tm3 responses are suppressed and sped up to a higher degree than the responses of Mi1 neurons (Fig 3). As these two cells constitute a large portion of T4 input [17, 38], the unequal effects of contrast normalization, depending on the presence of a local or a global stimulus, would skew the ratio of the neurons' outputs onto T4. To emphasize, this has severe implications for the resulting output of the computation because the composition of the inputs varies, depending on the profile of the visual stimulus.

## Mechanism of contrast normalization

The contrast normalization mechanism relies on neural feedback. At least part of this feedback comes from one or more medulla neurons [19]. The feedback does not solely rely on the output of the neuron itself, as blocking the cell's output is not enough to completely abolish contrast normalization of its response.

Even though the normalizing cell has not yet been identified, the search can be focused, as we can determine a number of characteristics that describe the feedback cell. Firstly, the normalizing feedback neuron has temporal band-pass filtering properties, as a stimulus with a high-contrast but static surround elicits the same response as a purely local stimulus (Fig 2). Secondly, the normalizing feedback cell is not selective for the direction of motion, as a surround with purely temporal dynamics is enough to reproduce the normalizing effects of a global stimulus that contains spatial motion. Thirdly, the normalizing feedback neuron either has a large receptive field or is a part of an interconnected network of the same cell type with smaller receptive fields, as the strength of the normalization increases as the surround grows in size, extending well beyond 50˚ [19].

We also make a clear prediction about the mechanism, via which the normalization is achieved in the neuron that exhibits the contrast normalization properties. The effects of contrast normalization on amplitude and kinetics of the signal are not a result of a simple saturation (Fig 4), instead, we can reproduce these effects by altering the cell's input resistance. We hypothesize that the input resistance of the cell should drop significantly in the presence of a global stimulus with a dynamic surround. However, measuring such a drop in the input resistance in the presence of the global dynamic surround at varying speeds might be a complicated experiment to perform. The contrast normalization effect described here is seen when imaging at the axon terminals of the cells, at the site of their synaptic output onto the first direction-selective cells, i.e. T4 and T5 cells. In the ON pathway, this corresponds to medulla layer M10. Input resistance measurements of these cells, however, are only possible at the soma, which, in the case of ON-pathway interneurons, is located outside of the medulla. Supposing that the neuronal computations are compartmentalized and local to the axons and dendrites, the drop in the input resistance might not reach the soma and thus not be measurable there.

To uncover the physiological mechanism underlying contrast normalization, one could start by focusing on octopamine, as the application of the octopamine receptor agonist chlordimeform (CDM) has been shown to significantly speed up response of *Drosophila* visual interneurons [14], and stimulus-dependence has been demonstrated to elicit changes in the shape of the response similar to those produced by octopamine [27]. The effect we observed here might be of a similar nature, mimicking the shift toward higher frequencies that occurs in the cells when the fly is in active locomotion [21], a state, in which it receives an abundance of global visual cues from the environment. Exploring the connection between the activation of octopamine receptors and the drop in the input resistance of the neuron in the presence of a global dynamic stimulus might shed further light on the mechanism of contrast normalization.

In summary, we built on the work in contrast normalization in the motion vision circuitry of *Drosophila melanogaster* and demonstrated the dramatic effect that contrast normalization has on the temporal characteristics of the interneurons' responses. Our findings illustrate the limitations of using simplified artificial stimuli with varying spatial profiles to probe the filtering properties of single cells and the constraints in transferring these results onto naturalistic conditions.

## Supporting information

**S1 Fig. Dynamic surround affects response amplitude and kinetics.** Mi1 responses to a luminance step in the RF center with gray, stationary, moving grating, and stochastic stimulus surround; n = 48 cells/11 flies. Luminance step occurred during the gray-shaded period. Left: Amplitudes of cell responses. Shaded areas around the curves show bootstrapped 68% confidence intervals. Right: Kinetics of cell responses. Responses during each condition are normalized to the condition's maximum.
(TIF)

**S2 Fig. Dynamic surround and input amplitude do not affect L1 response kinetics.** (A) Schematic representation of early stages of the motion detection circuit. Highlighted is a lamina neuron in the ON pathway that doesn't exhibit contrast normalization properties and provides major input to contrast normalization-exhibiting neurons. (B) Contrast tuning curves for lamina neuron L1 (n = 14 cells/4 flies). Shaded areas around the curves show bootstrapped 68% confidence intervals. Adapted from Drews et al. (2020) [19]. (C) Spatial and temporal stimulus profile. Left: 3 luminance step amplitudes with gray surround. Right: 3 luminance step amplitudes with moving grating surround. (D) L1 responses to luminance steps of different amplitudes with gray and moving grating surround. Luminance step happened during gray-shaded period. Left: Amplitudes of cell responses. Shaded areas around the curves show bootstrapped 68% confidence intervals. Right: Response kinetics. Responses during each condition are normalized to condition's maximum.
(TIF)

**S3 Fig. Visual surround dynamics significantly suppress signal amplitude and increase its kinetics.** (A-I) Left: Maximum signal amplitude reached in each condition as a percentage of peak signal amplitude reached in gray surround condition. Right: Level of signal decay reached by the end of the luminance step (between 0.9 and 1.1 seconds from the start of stimulation) in each condition, as a percentage of the peak signal amplitude reached in that condition. (A) Statistics for Tm3 responses in Fig 2. Left: Amplitudes of cell responses. In comparison to the gray surround condition, response amplitude is the same in stationary grating surround condition (Mann-Whitney U: 2357, NS p = 0.46), and is suppressed when a moving grating (Mann-Whitney U: 121, ***p < 0.001) or a stochastic stimulus (Mann-Whitney U: 535, ***p < 0.001) is presented in the surround. Right: Kinetics of cell responses. In comparison to the gray surround condition, the signal decays to the same level in the stationary grating surround condition (Mann-Whitney U: 2347, NS p = 0.44), and decays significantly stronger if a moving grating (Mann-Whitney U: 777, ***p < 0.001) or a stochastic stimulus (Mann-Whitney U: 499, ***p < 0.001) is presented in the surround. (B) Statistics for Mi1 responses in S1 Fig. Left: Amplitudes of cell responses. In comparison to the gray surround condition, the amplitude of the signal is the same in stationary grating surround condition (Mann-Whitney U: 10345, NS p = 0.49), and is suppressed when a moving grating (Mann-Whitney U: 6210, ***p < 0.001) or a stochastic stimulus (Mann-Whitney U: 6223, ***p < 0.001) is presented in the surround. Right: Kinetics of cell responses. In comparison to the gray surround condition, the signal decays to the same level in the stationary grating surround condition (Mann-Whitney U: 10266, NS p = 0.44), and decays stronger if a moving grating (Mann-Whitney U: 8101, ***p < 0.001) or a stochastic stimulus (Mann-Whitney U: 8545, **p = 0.005) is presented in the surround. (C) Statistics for Mi1 responses in Fig 3. Left: Amplitudes of cell responses. In comparison to the gray surround condition, the amplitude of the signal is suppressed when a moving grating (Mann-Whitney U: 24770, ***p < 0.001) is presented in the surround. Right: Kinetics of cell responses. In comparison to the gray surround condition, the signal decays stronger if a moving grating (Mann-Whitney U: 34725, ***p < 0.001) is presented in the surround. (D) Statistics for Tm1 responses in Fig 3. Left: Amplitudes of cell responses. In comparison to the gray surround condition, the amplitude of the signal is suppressed when a moving grating (Mann-Whitney U: 1065, ***p < 0.001) is presented in the surround. Right: Kinetics of cell responses. In comparison to the gray surround condition, the signal decays stronger if a moving grating (Mann-Whitney U: 1954, **p = 0.005) is presented in the surround. (E) Statistics for Tm2 responses in Fig 3. Left: Amplitudes of cell responses. In comparison to the gray surround condition, the amplitude of the signal is suppressed when a moving grating (Mann-Whitney U: 900, ***p < 0.001) is presented in the surround. Right: Kinetics of cell responses.

In comparison to the gray surround condition, the signal decays stronger if a moving grating (Mann-Whitney U: 1578, **p = 0.003) is presented in the surround. (F) Statistics for Tm3 responses in Fig 3. Left: Amplitudes of cell responses. In comparison to the gray surround condition, the amplitude of the signal is suppressed when a moving grating (Mann-Whitney U: 1138, ***p < 0.001) was presented in the surround. Right: Kinetics of cell responses. In comparison to the gray surround condition, the signal decays stronger if a moving grating (Mann-Whitney U: 6433, ***p < 0.001) is presented in the surround. (G) Statistics for Tm3 responses in Fig 5. Left: Amplitudes of cell responses. For the gray surround condition, in comparison to the response to 100% luminance step, the amplitude of the signal is the same in response to 75% luminance step (Mann-Whitney U: 1668, NS p = 0.06) and is suppressed in response to 50% luminance step (Mann-Whitney U: 1668, ***p < 0.001). In comparison to the response to 100% luminance step in gray surround condition, the amplitude of the signal is suppressed in the moving grating surround condition in response to 100% (Mann-Whitney U: 273, ***p < 0.001), 75% (Mann-Whitney U: 38, ***p < 0.001), and 50% (Mann-Whitney U: 4, ***p < 0.001) luminance steps. In the moving grating surround condition, in comparison to the response to 100% luminance step, the amplitude of the signal is suppressed in response to 75% (Mann-Whitney U: 1512, *p = 0.011) and 50% (Mann-Whitney U: 995, ***p < 0.001) luminance steps. Right: Kinetics of cell responses. In gray surround condition, in comparison to the response to 100% luminance step, the signal decays to the same level in response to 75% (Mann-Whitney U: 1963, NS p = 0.46) and 50% (Mann-Whitney U: 1712, NS p = 0.09) luminance steps. In comparison to the 100% luminance step in gray surround condition, the signal decays stronger in moving grating surround condition in response to 100% (Mann-Whitney U: 957, ***p < 0.001), 75% (Mann-Whitney U: 789, ***p < 0.001), and 50% (Mann-Whitney U: 713, ***p < 0.001) luminance steps. In the moving grating surround condition, in comparison to the 100% luminance step, the signal decays to the same level in response to 75% (Mann-Whitney U: 1747, NS p = 0.12) and 50% (Mann-Whitney U: 1769, NS p = 0.44) luminance steps. (H) Statistics for Mi1 responses in Fig 5. Left: Amplitudes of cell responses. In gray surround condition, in comparison to 100% luminance step, the amplitude of the signal is the same in response to 75% (Mann-Whitney U: 2955, NS p = 0.38) and is suppressed in response to 50% (Mann-Whitney U: 2321, **p = 0.005) luminance steps. In comparison to 100% luminance step in gray surround condition, the amplitude of the signal is suppressed in moving grating surround condition in response to 100% (Mann-Whitney U: 1582, ***p < 0.001), 75% (Mann-Whitney U: 1056, ***p < 0.001), and 50% (Mann-Whitney U: 633, ***p < 0.001) luminance steps. In the moving grating surround condition, in comparison to 100% luminance step, the amplitude of the signal is suppressed in response to 75% (Mann-Whitney U: 2355, **p = 0.007) and 50% (Mann-Whitney U: 1617, ***p < 0.001) luminance step. Right: Kinetics of cell responses. In gray surround condition, in comparison to 100% luminance step, the signal decays to the same level in response to 75% (Mann-Whitney U: 3042, NS p = 0.5) and 50% (Mann-Whitney U: 3035, NS p = 0.5) luminance steps. In comparison to 100% luminance step in gray surround condition, the signal decays to the same level in moving grating surround condition in response to 75% luminance step (Mann-Whitney U: 2603, NS p = 0.06), and decays stronger in response to 100% (Mann-Whitney U: 2539, *p = 0.037) and 50% (Mann-Whitney U: 2370, **p = 0.009) luminance steps. In the moving grating surround condition, in comparison to 100% luminance step, the signal decays to the same level in response to 75% (Mann-Whitney U: 3034, NS p = 0.5) and 50% (Mann-Whitney U: 2720, NS p = 0.13) luminance steps. (I) Statistics for L1 responses in S2 Fig. Left: Amplitudes of cell responses. In the gray surround condition, in comparison to 100% luminance step, the amplitude of the signal is suppressed in response to 75% (Mann-Whitney U: 572, **p = 0.003) and 50% (Mann-Whitney U: 381, ***p < 0.001) luminance steps. In comparison to 100% luminance step in the gray

surround condition, the amplitude of the signal in moving grating surround condition is the same in response to 100% luminance step (Mann-Whitney U: 778, NS p = 0.18), and is suppressed in response to 75% (Mann-Whitney U: 609, **p < 0.007), and 50% (Mann-Whitney U: 393, ***p < 0.001) luminance steps. The amplitude of the signal is the same in response to 75% luminance step in the gray surround condition and in the moving grating surround condition (Mann-Whitney U: 867, NS p = 0.45). The amplitude of the signal is the same in response to 50% luminance step in the gray surround condition and in the moving grating surround condition (Mann-Whitney U: 828, NS p = 0.32). Right: Kinetics of cell responses. In comparison to 100% luminance step in gray surround condition, the signal decays to the same level in the gray surround condition in response to 75% (Mann-Whitney U: 853, NS p = 0.4) and 50% (Mann-Whitney U: 801, NS p = 0.24) luminance steps, as well as in the moving grating surround condition in response to 100% (Mann-Whitney U: 874, NS p = 0.47), 75% (Mann-Whitney U: 812, NS p = 0.27), and 50% (Mann-Whitney U: 783, NS p = 0.19) luminance step.
(TIF)

**S4 Fig. Visual surround outside cell's RF does not elicit a response from the cell.** Extended traces of Fig 3 average responses of contrast normalization-exhibiting neurons to luminance step in the RF center with gray and moving grating surround. Mi1 (n = 98 cells/25 flies); Tm1 (n = 24 cells/9 flies); Tm2 (n = 22 cells/7 flies), and Tm3 (n = 65 cells/16 flies). Luminance step happened during the gray-shaded period. The dashed lines denote the time when the surround was on screen. Shaded areas around the curves show bootstrapped 68% confidence intervals.
(TIF)

## Acknowledgments

We thank Jurgen Haag and Lukas Groschner for commenting on drafts of the manuscript. We thank Wolfgang Essbauer and Michael Sauter for fly husbandry. N.P. and A.B. are members of the Graduate School of Systemic Neurosciences (GSN) Munich.

## Author Contributions

**Conceptualization:** Nadezhda Pirogova, Alexander Borst.

**Data curation:** Nadezhda Pirogova.

**Formal analysis:** Nadezhda Pirogova, Alexander Borst.

**Funding acquisition:** Alexander Borst.

**Investigation:** Nadezhda Pirogova.

**Methodology:** Nadezhda Pirogova, Alexander Borst.

**Project administration:** Nadezhda Pirogova.

**Supervision:** Alexander Borst.

**Validation:** Nadezhda Pirogova.

**Visualization:** Nadezhda Pirogova.

**Writing – original draft:** Nadezhda Pirogova.

**Writing – review & editing:** Alexander Borst.

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
