## [Decision Letter · Decision Letter 0]

17 Jan 2023

PONE-D-22-30015Contrast Normalization Affects Response Time-Course of Visual InterneuronsPLOS ONE

Dear Dr. Pirogova,

Thank you for submitting your manuscript to PLOS ONE. After careful consideration, we feel that it has merit but does not fully meet PLOS ONE’s publication criteria as it currently stands. Therefore, we invite you to submit a revised version of the manuscript that addresses the points raised during the review process.

Both reviewers had similar comments about the submitted manuscript that should be addressed. Specifically, please provide additional statistical analyses and clarification of some of the analyses done. Please provide the additional background and context raised by both reviewers.

We look forward to receiving your revised manuscript.

Kind regards,

Melissa J. Coleman

Academic Editor

PLOS ONE

Journal Requirements:

Reviewers' comments:

Reviewer's Responses to Questions

**Comments to the Author**

1. Is the manuscript technically sound, and do the data support the conclusions?

Reviewer #1: Partly

Reviewer #2: Yes

2. Has the statistical analysis been performed appropriately and rigorously? 

Reviewer #1: No

Reviewer #2: Yes

3. Have the authors made all data underlying the findings in their manuscript fully available?

Reviewer #1: Yes

Reviewer #2: No

4. Is the manuscript presented in an intelligible fashion and written in standard English?

Reviewer #1: Yes

Reviewer #2: Yes

5. Review Comments to the Author

Reviewer #1: Major comments

1) The introduction to the paper is missing previous work that has been done both in larger flies (e.g. Harris Neuron 2000) and in Drosophila (e.g. recent papers from the Silies lab). Additionally, the Matulis 2019 should be mentioned already in the introduction as part of the background for the current work.

2) The paper is missing statistical tests. All claims regarding changes in amplitude and kinetics should be explicitly tested with the appropriate statistical test (e.g. Drews 2020 paper, Fig 5).

3) Authors’ choice of variability measure throughout the paper is a bit unorthodox (68% bootstrapped confidence interval). If authors choose to continue to use it, they should both justify the selection and present at least some of their results in the supplementary material with a more conventional statistics of variability.

4) Authors should present the entire stimulus trace, from before the surround presentation to after its disappearance. This should be done at least once for each cell type so that it will be clear to readers that the surround is indeed a classic surround that does not evoke a response.

Comments

1) State explicitly that the numbers in the figure legends are for neurons and flies.

2) In Figures 2,3 and 5, the figure legend claims that responses are normalized to the cell’s maximum, but the y-axis seems to represent a non-normalized response (simple df/f).

3) Please clarify if the Tm3 dataset in Fig 3 contains the neurons from fig 2 or if it is completely independent.

4) Please address the large difference in the numbers of cells imaged for the different cell types (i.e. 98 Mi1s vs. 22 Tm2)

5) The Matulis 2020 paper does contain local stimuli as part of their stimulus set. The explanation for the discrepancy in the discussion should be corrected.

6) Included code for figure replication is using a function that has been removed since 2020 (tsplot - https://seaborn.pydata.org/whatsnew/v0.10.0.html?highlight=tsplot). Please correct to updated versions of python and associated packages.

7) Line 170 states that "on average, over 90% of all cells measured passed the SNR criterion". Please clarify the sentence.

Reviewer #2: The manuscript by Pirogova and Borst describes contrast normalization in several medulla neurons (Mi1, Tm3, Tm1, Tm2) that feed into the motion-sensitive direction-selective T4 and T5 neurons in Drosophila. While contrast normalization in these medulla neurons (specifically, the effects on response amplitudes) has been described by Drew et al., 2020, the present study demonstrated the effects on temporal properties or response dynamics, exerted only by dynamic (but not stationary) stimuli in the visual surround. To account for these effects, the authors proposed two models, one with a stationary nonlinearity and the other with dynamic nonlinearity. By showing that the temporal properties of the responses was affected by the stimulus surround and not simply consequential to response saturation, the authors excluded the first model and favored the dynamic nonlinearity model with the hypothesis that contrast normalization affects membrane time constant.

Overall, the results represent an interesting extension of Drew et al., 2020. The data are of high quality. I have a number of minor comments/suggestions but would otherwise support the publication of this manuscript.

(1) Temporal dynamic is the focus of this study but was only presented in a qualitative way. It would be very helpful if the authors could quantify the effects or parameterize the temporal dynamics, and potentially compare with the modeling results.

(2) I am not sure if the presentation of the stationary nonlinearity model is really necessary. On the other hand, must the dynamic nonlinearity model utilize the alteration of input resistance ? I suppose not. The comparison of these two very different types of models seems odd. The authors could do a better job to introduce the models.

(3) The receptive field sizes for the medulla neurons (described in 241-243) appear to be different from those described in Arenz et al., 2017.

6. PLOS authors have the option to publish the peer review history of their article (what does this mean?). If published, this will include your full peer review and any attached files.

Reviewer #1: No

Reviewer #2: No

---

## [Author Response · Author response to Decision Letter 0]

15 Mar 2023

Dear Editor,

Thank you for considering our manuscript entitled ‘Contrast Normalization Affects Response Time-Course of Visual Interneurons’ for publication. We appreciate the reviewers' thoughtful comments and suggestions on our work. We have addressed each of the reviewers' comments below.

Reviewer #1: Major comments

1) The introduction to the paper is missing previous work that has been done both in larger flies (e.g. Harris Neuron 2000) and in Drosophila (e.g. recent papers from the Silies lab). Additionally, the Matulis 2019 should be mentioned already in the introduction as part of the background for the current work.

We added the citations to previous work [37] Harris et al. (2000) and [38] Ketkar et al. (2022) in the introduction and corrected the introduction to include Matulis et al. (2020) as part of the background.

2) The paper is missing statistical tests. All claims regarding changes in amplitude and kinetics should be explicitly tested with the appropriate statistical test (e.g. Drews 2020 paper, Fig 5).

We have added a figure (S3) that shows the results of Mann-Whitney U test (as in Drews et al., 2020) for each of the existing figures, included the indications of significance in the manuscript, and referred to figure S3 in the text. 

3) Authors’ choice of variability measure throughout the paper is a bit unorthodox (68% bootstrapped confidence interval). If authors choose to continue to use it, they should both justify the selection and present at least some of their results in the supplementary material with a more conventional statistics of variability.

We use 68% bootstrapped confidence interval in the figures when the whole trace of the response is presented, as in Drews et al. (2020). We have now added figure S3 with additional statistical analysis for each of the main and supplementary figures that show 68% bootstrapped confidence interval as a measure of variability. 

4) Authors should present the entire stimulus trace, from before the surround presentation to after its disappearance. This should be done at least once for each cell type so that it will be clear to readers that the surround is indeed a classic surround that does not evoke a response.

We have added a figure (S4) that shows extended traces of the responses from Fig. 3 for each cell type and referred to figure S4 in the text.

Comments

1) State explicitly that the numbers in the figure legends are for neurons and flies.

The legends have been corrected to explicitly state the number of cells and flies in the dataset.

2) In Figures 2,3 and 5, the figure legend claims that responses are normalized to the cell’s maximum, but the y-axis seems to represent a non-normalized response (simple df/f).

We apologise for the mistake. We have corrected the legends accordingly.

3) Please clarify if the Tm3 dataset in Fig 3 contains the neurons from fig 2 or if it is completely independent.

The Tm3 and Mi1 dataset in Fig 3 contains the data from Fig 2 and Fig 3. The legend for Fig 3 was corrected accordingly.

4) Please address the large difference in the numbers of cells imaged for the different cell types (i.e. 98 Mi1s vs. 22 Tm2)

Mi1 was imaged with a number of protocols that had different combinations of stimuli that all contained the ‘grey surround’ condition and the ‘moving grating’ condition presented in Fig 3. This resulted in a smaller number of Mi1 cells stimulated with a stochastic surround that was only in one stimulus set, but a higher number of responses (from all the imaged Mi1 cells) to the two conditions shown in Fig 3. These different stimulus protocols were combined into one when also other cell types were recorded, resulting in a lower number of the other cells imaged.

5) The Matulis 2020 paper does contain local stimuli as part of their stimulus set. The explanation for the discrepancy in the discussion should be corrected.

We apologise for the mistake. We have corrected the explanation accordingly.

6) Included code for figure replication is using a function that has been removed since 2020 (tsplot - https://seaborn.pydata.org/whatsnew/v0.10.0.html?highlight=tsplot). Please correct to updated versions of python and associated packages.

We have now provided a yaml file in the GitHub repository to recreate a Python environment that contains all the Python packages needed to reproduce the figures.

7) Line 170 states that "on average, over 90% of all cells measured passed the SNR criterion". Please clarify the sentence.

To pass the SNR criterion to become included in the data, the inter-trial variance of a cell’s responses had to be smaller than the average cell response. Large inter-trial variance was typically caused by movement artefacts. Averaged over all cell types recorded, less than 10% of the cells were discarded due to the SNR criterion.

Reviewer #2 Minor comments/suggestions:

(1) Temporal dynamic is the focus of this study but was only presented in a qualitative way. It would be very helpful if the authors could quantify the effects or parameterize the temporal dynamics, and potentially compare with the modeling results.

We parameterized and quantified the temporal dynamics of the response by measuring the level to which the response has decayed at the end of the luminance step as percent of the peak responses. We have now also added a figure (S3) that shows the results of Mann-Whitney U test (as in Drews et al., 2020) for each of the existing figures, included the indications of significance in the manuscript, and referred to figure S3 in the text. 

(2) I am not sure if the presentation of the stationary nonlinearity model is really necessary. On the other hand, must the dynamic nonlinearity model utilize the alteration of input resistance ? I suppose not. The comparison of these two very different types of models seems odd. The authors could do a better job to introduce the models.

We still think that our attempt to explain the phenomenon with a stationary nonlinearity first makes sense, since this is the classical way to describe contrast normalization. Furthermore, even this stationary saturation nonlinearity affects the time-course of the signal due to signal compression for large amplitudes. We actually learned a lot from it: if this is the explanation, we should obtain the same signal amplitude and time course for a static surround that we observed for a moving surround by simply reducing the stimulus amplitude accordingly. Clearly, the results contradicted this explanation. On the other hand, the biophysical model could reproduce the experimental results well. 

To be clearer on this, we now explicitly introduce an inhibitory input which acts as a shunt, instead of saying that the leak conductance is enlarged by the moving surround. 

(3) The receptive field sizes for the medulla neurons (described in 241-243) appear to be different from those described in Arenz et al., 2017.

The numbers are taken from S1 and S2 tables in Arenz et. al., 2017 (Mi1 - 28.81°, Tm3 - 11.91°, Tm1 - 27.14°, Tm2 - 30.52°) and rounded to the nearest integer, resulting in the values used (29° for Mi1 cells, 12° for Tm3 cells, 27° for Tm1 cells, and 31° for Tm2 cells). 

We clarified the way the values were rounded in the manuscript.

---

## [Decision Letter · Decision Letter 1]

28 Apr 2023

Contrast Normalization Affects Response Time-Course of Visual Interneurons

PONE-D-22-30015R1

Dear Dr. Pirogova,

We’re pleased to inform you that your manuscript has been judged scientifically suitable for publication and will be formally accepted for publication once it meets all outstanding technical requirements.

Kind regards,

Melissa J. Coleman

Academic Editor

PLOS ONE

Additional Editor Comments (optional):

Reviewers' comments:

Reviewer's Responses to Questions

**Comments to the Author**

1. If the authors have adequately addressed your comments raised in a previous round of review and you feel that this manuscript is now acceptable for publication, you may indicate that here to bypass the “Comments to the Author” section, enter your conflict of interest statement in the “Confidential to Editor” section, and submit your "Accept" recommendation.

Reviewer #1: All comments have been addressed

2. Is the manuscript technically sound, and do the data support the conclusions?

Reviewer #1: Yes

3. Has the statistical analysis been performed appropriately and rigorously? 

Reviewer #1: Yes

4. Have the authors made all data underlying the findings in their manuscript fully available?

Reviewer #1: Yes

5. Is the manuscript presented in an intelligible fashion and written in standard English?

Reviewer #1: Yes

6. Review Comments to the Author

Reviewer #1: (No Response)

7. PLOS authors have the option to publish the peer review history of their article (what does this mean?). If published, this will include your full peer review and any attached files.

Reviewer #1: No

---

## [Editor Report · Acceptance letter]

1 Jun 2023

PONE-D-22-30015R1 

Contrast Normalization Affects Response Time-Course of Visual Interneurons 

Dear Dr. Pirogova:

I'm pleased to inform you that your manuscript has been deemed suitable for publication in PLOS ONE. Congratulations! Your manuscript is now with our production department. 

Kind regards, 

on behalf of

Dr. Melissa J. Coleman 

Academic Editor

PLOS ONE